# The Association between Biofilm Formation and Antimicrobial Resistance with Possible Ingenious Bio-Remedial Approaches

**DOI:** 10.3390/antibiotics11070930

**Published:** 2022-07-11

**Authors:** Yogesh Dutt, Ruby Dhiman, Tanya Singh, Arpana Vibhuti, Archana Gupta, Ramendra Pati Pandey, V. Samuel Raj, Chung-Ming Chang, Anjali Priyadarshini

**Affiliations:** 1Department of Microbiology, SRM University, Rajiv Gandhi Education City, Post Office P.S. Rai, Sonepat 131029, Haryana, India; yogeshkind@gmail.com (Y.D.); rubydhiman17@gmail.com (R.D.); arpana.v@srmuniversity.ac.in (A.V.); archana.g@srmuniversity.ac.in (A.G.); ramendra.pandey@gmail.com (R.P.P.); directorcd4@srmuniversity.ac.in (V.S.R.); 2Department of Botany, TPS College, Patliputra University, Patna 800020, Bihar, India; tanyasingh10aug@gmail.com; 3Master & Ph.D. Program in Biotechnology Industry, Chang Gung University, No.259, Wenhua 1st Rd., Guishan Dist., Taoyuan City 33302, Taiwan

**Keywords:** biofilm, AMR, silver nanoparticles, multidrug resistance, extracellular polymeric substances, biofilm control

## Abstract

Biofilm has garnered a lot of interest due to concerns in various sectors such as public health, medicine, and the pharmaceutical industry. Biofilm-producing bacteria show a remarkable drug resistance capability, leading to an increase in morbidity and mortality. This results in enormous economic pressure on the healthcare sector. The development of biofilms is a complex phenomenon governed by multiple factors. Several attempts have been made to unravel the events of biofilm formation; and, such efforts have provided insights into the mechanisms to target for the therapy. Owing to the fact that the biofilm-state makes the bacterial pathogens significantly resistant to antibiotics, targeting pathogens within biofilm is indeed a lucrative prospect. The available drugs can be repurposed to eradicate the pathogen, and as a result, ease the antimicrobial treatment burden. Biofilm formers and their infections have also been found in plants, livestock, and humans. The advent of novel strategies such as bioinformatics tools in treating, as well as preventing, biofilm formation has gained a great deal of attention. Development of newfangled anti-biofilm agents, such as silver nanoparticles, may be accomplished through omics approaches such as transcriptomics, metabolomics, and proteomics. Nanoparticles’ anti-biofilm properties could help to reduce antimicrobial resistance (AMR). This approach may also be integrated for a better understanding of biofilm biology, guided by mechanistic understanding, virtual screening, and machine learning in silico techniques for discovering small molecules in order to inhibit key biofilm regulators. This stimulated research is a rapidly growing field for applicable control measures to prevent biofilm formation. Therefore, the current article discusses the current understanding of biofilm formation, antibiotic resistance mechanisms in bacterial biofilm, and the novel therapeutic strategies to combat biofilm-mediated infections.

## 1. Introduction

The existence of microbial pathogens has been recognized for many years, and since then, researchers have continuously tried to eliminate already existing and emerging pathogens causing infectious diseases, and develop antimicrobial agents to treat and eliminate the infectious disease. Antimicrobials represent a broad spectrum of compounds that may act against a wide repertoire of disease-causing microorganisms, such as bacteria, viruses, parasites, fungi, and protozoa [1]. These compounds have been used since the early 20th century to treat infected patients and have helped significantly in lowering the morbidity as well as mortality rates of most infectious diseases. In 1928, Alexander Fleming discovered penicillin, and in the 1940s, it came into clinical use just in time for World War II [2]. After only four years of use, the first penicillin-resistant strains of bacteria emerged; this resulted in the evolution of antimicrobial resistance (AMR). Subsequently, AMR has accelerated rapidly and expanded to different pathogenic species due to the persistent exposure and non-targeted application of antimicrobials in clinical and agricultural settings. Many current antimicrobials have become ineffective due to the advancement of AMR; it is believed that the non-judicious use and overdosing of antimicrobials are one of the foremost factors for the rise in drug-resistant bacteria [3]. Over last few decades, researchers have been trying to understand the mechanism of AMR and to develop new antimicrobials.

The ability to form biofilms is a universal attribute of bacteria. Biofilms are multicellular communities held together by a self-produced extracellular matrix. The mechanisms that a number of bacteria employ to form biofilms vary, frequently depending on environmental conditions and specific strain attributes. In the 17th century, Antonie van Leeuwenhoek first observed “animalcules” on his own teeth. In 1940, the “bottle effect” was observed in marine microorganisms [4]. This showed that the bacteria grow more often on the surface. Then, in 1943, biofilms were made by Zobell, and he found that bacteria on surfaces were greater in number compared with the surrounding seawater [5].

Today, we generally define such biofilms as microbial communities adhered to a substratum and encased within an extracellular polymeric substance (EPS) produced by the microbial cells themselves [6,7]. Microbial biofilms are found on the surfaces of medical devices, dental implants, suturing materials, catheters, and human and animal tissues; also in aquatic mediums, damp structures, natural and artificial environmental conditions, and plant roots in the disease-causing forms with the ability to release toxins to the surrounding matrix [4,7]. Apart from that, biofilms also occur in human or animal alimentary canals, wastewater filters, or aquatic bodies in symbiotic form [8]. Biofilms are most prevalent in natural environments; and are responsible for causing infections in humans and animals due to their resistance to antimicrobials [9,10]. Therefore, it is very much pertinent to fully understand biofilm-led survival against antibiotics [11]. In this review, we will discuss the AMR, mechanism of biofilm-led AMR, and small molecules and drug candidates for the potential anti-biofilm therapies.

## 2. Overview of Biofilm-Led Antibiotic Survival

Biofilm is a consortium of microorganisms in which cells adhere to each other and often to almost every surface; it can form one layer in direct contact with the substratum or form in flocs, is mobile, and can form complex communities (Figure 1).

These adherent cells grow in multicellular aggregates and are embedded in a matrix composed of extracellular polymeric substances (EPS). New properties usually emerge in the biofilm, which are clearly distinct from the bacterial life and could not be predicted from the bacterial life. Biofilms are most widely distributed in water, soil, sediment, and subsurface environment, and navigate the biogeochemical cycling processes of most elements. Microorganisms colonize all higher organisms, including humans, and form biofilms [12]. Biofilm can form on living as well as non-living surfaces such as prosthetic or artificial teeth (Figure 2e,f), and may be prevalent in nature, industries, and hospitals. For instance, biofilm can form as early soft dental plaque or more mature mineralized and calcified plaque (calculus) on human teeth (Figure 2), skin, and urinary tract; also on medical devices such as catheters and implants that can result in chronic infections [4,13]. Moreover, they are responsible for contamination of process water, as well as biofouling, and they worsen the hygienic quality of drinking water [12,14].

Biofilm can accommodate several bacterial species (Figure 1), form complex systems, and can have high cell densities ranging from 10^8^ to 10^11^ cells g^−1^ wet weight [15]. Biofilms form a heterogenous ecosystem as soon as their cells undergo differentiation and synchronize their life cycles. Novel structures, activities, patterns, and properties that arise during the process, and of self-organization, are the emergent properties of biofilm communities (Figure 1). The main component of the matrix is water (up to 97%). Structural and functional components of the matrix include soluble, gel-forming polysaccharides, proteins, and extracellular DNA (eDNA), as well as insoluble components, such as amyloids, cellulose, fimbriae, pili, and flagellae. As the biofilm formation is a dynamic process, intermolecular interactions between EPS components produce new EPS molecules in the matrix that determine the physiological activity and mechanical properties of the matrix of the organism, and the biofilm architecture is solely because of these EPS molecules [16]. Pores and channels between microcolonies form the voids in the matrix that ease the liquid transport. Extracellular DNA forms a stable filamentous network structure (Figure 3). Tolerance to desiccation is also an emergent property of biofilm conferred by the matrix [17].

Bacterial biofilms are one of the key factors in chronic infections on the grounds of higher tolerance to antibiotics and disinfectants; they can combat phagocytosis and other components of the immune system [18]. Consequently, microorganisms in biofilms become less susceptible to multiple antimicrobial agents, which drives biofilms to an impending predicament in therapeutics [19]. Nonetheless, antibiotic tolerance is predominantly contingent on the formation of biofilm, the composition of ECM with proteins, lipids, water, glycolipids, polysaccharides, surfactants, extracellular DNA (eDNA), membrane vesicles, and extracellular RNA, and the architecture of biofilm, which refers to the biomass and space organization within the biofilm [16,20,21,22].

## 3. Mechanism of Biofilm Resistance

Antimicrobials (antibiotics, antivirals, antifungals, and antiparasitics) are compounds that kill microbes, stop their growth, and prevent or treat the infections in humans, animals, and plants. Antimicrobial resistance is the ability of microorganisms to survive against an antimicrobial drug at a higher concentration for a longer period, and is measured as minimum inhibitory concentration (MIC) [23]. Biofilm resistance can be antibiotic resistance or antibiotic tolerance. Microorganisms develop mechanisms against antimicrobials either through acquisition of foreign genetic material coding for resistant determinants by horizontal gene transfer (HGT) in biofilm EPS, or through genetic mutation (Figure 3). Mechanisms of AMR are: prevention of access or reduced permeability of antimicrobials, mutational changes in antibiotic targets, modification of targets, and enzymatic degradation of the antimicrobials by hydrolysis or chemical change (Figure 4). Antibiotic resistance (ABR) is a subdivision of AMR, as antibiotics are effective against the bacteria, but the bacteria become resistant to antibiotics.

Antimicrobial resistance can be intrinsic (naturally acquired) by genetic mutation, genes encoding inherent structural and functional traits of the molecular target, acquired extrinsically, or can be adaptive. Intrinsically, antibiotics such as vancomycin and daptomycin, which are active against Gram-positive bacteria, might not be effective against the Gram-negative bacteria due to distinct cell wall composition. On the other hand, acquired resistance involves genetic modification either through HGT or mutation. The bacteria can also adapt the capacity of resistance via gene expression and protein production rapidly in response to stress or other environmental conditions, and also in the presence of specific antibiotics. Thioredoxin A (*Ttrx A*), thioredoxin reductase (*trxB*), D-Ala-D-Ala carboxypeptidase, *DacA*, *FabI*, and *SapC* are a few resistance genes which are responsible for the innate resistance to antibiotics such as fluoroquinolones, β-lactams, and aminoglycosides [24].

However, tolerance is the ability of microorganisms to withstand or survive antibiotics higher than the inhibitory concentration for a period of time [25]. Tolerance is an adaptive mechanism that reflects the change in cellular behavior from an active state to a dormant or inactive state (Figure 3) for a transient period [26]. A major rearrangement of energy production or few miscellaneous functions are witnessed during the tolerant state, and can be seen during poor growth or in the presence of few antibiotics or stress. Entrapment of antibiotics to the ECM without attachment to the target can also trigger tolerance, and results in dormancy or non-growth of bacterial cells. Persistence is a phenomenal form of tolerance, and persisters (Figure 3) are the tolerant form of cells in the population that are capable of surviving antibiotics but can be killed with longer exposure [27]. They can be either triggered persisters, induced in the presence of environmental stress or condition, or spontaneous persisters, converted to the form after a stochastic process. Persistence is also called heterotolerance [28].

Biofilm-mediated resistance (Figure 3) is a complex form of resistance that requires both the mechanisms of antibiotic resistance, as well as tolerance. Additionally, bacterial strains and species, antimicrobial agents, condition and developmental state of biofilm, and the growth condition of biofilm can highly affect the overall process [25].

### 3.1. Prevention of Access or Reduced Penetration

The architecture, as well as composition, of ECM through gradients of dispersion can severely affect the penetration of antibiotics, the access to cells, and, finally, affect the efficacy of antibiotics. Diffusion of antibiotics also varies due to interaction with the ECM components [29]. For example, extracellular DNA enhances the resistance of *Pseudomonas* biofilm against aminoglycosides, but not against beta-lactams and fluoroquinolones [30,31,32]. In the same way, eDNA enhances the resistance of *Staphylococcus epidermidis* biofilm against glycopeptides. It has been seen that negatively charged eDNA binds to negatively charged glycopeptides (vancomycin) and aminoglycosides (tobramycin). It has also been demonstrated that the binding of vancomycin and eDNA is 100-fold higher than the vancomycin and D-Ala-D-Ala peptides in peptidoglycan precursors; this may result in the accumulation of eDNA in the ECM [32]. The retention of eDNA in the ECM may result in a cation-limited environment through the chelation of Mg^2+^ cations. The chelation of Mg^2+^ can also initiate the AMR-linked two-component system of PhoPQ and PmrAB for *Psuedomonas aeruginosa* and *S. enterica* serovar Typhimurium [31,33].

Additionally, the antibiotic modifying enzymes can be released and located in the ECM; these can also be used by other sensitive species of bacteria within a mixed species biofilm. For instance, beta lactamases released by *Moraxella catarrhalis* protect the *S. pneumoniae* and *H. influenza* against amoxicillin and ampicillin, respectively [34,35]. Therefore, the biofilm architecture can alter the diffusion of antibiotics, and also the exposure of cells.

### 3.2. Stress Responses and Nutritional Limitation

Physiological heterogeneity is characterized by the genetic and phenotypic expression, metabolic activity, and antibiotic tolerance between the cells within different areas of biofilm [36,37]. The organization and architecture of biofilm generates gradients (Figure 3) of dispersion of water, nutrients, pH, signaling molecules, and waste products. It is believed that cells near the surface of biofilm microcolony utilize most of the nutrient supplies and create a deprived area further down [38,39,40,41]. Development of oxygen and nutritional depletion (Figure 3) in microcolonies and lower niches can contribute to the development of diverse physiological states of aerobic, anaerobic, microaerobic, and fermentative conditions; also the development of persisters, slow growth, fast growth, and dormant cells [17,36]. One such unique feature was observed by Yogesh and Anjali in 2021 (unpublished data), when MDR *Enterococcus faecalis* strains were re-cultured directly from the biofilm stage, which were stored for an extended period of 16 to 18 months at −70 °C. They found that the re-cultured colonies could grow only after 60–72 h of incubation at 37 °C on nutrient or chromogenic UTI agar without any supplement. When examined by the investigators of this study, the re-cultured bacterial strains were found to be resistant for the additional antibiotics. Cells in the oxygen deprived area (Figure 3) can show reduced metabolic activities and may undergo dormancy; this is believed to be the reason for tolerance against antibiotics such as tobramycin and ciprofloxacin that target protein synthesis and DNA gyrase [42].

It is well established that slow glowing cells are susceptible to colistin, which acts on the cell membrane [43]. However, the presence of colistin-tolerant cells within oxygen rich areas was observed, suggesting disagreement with the connection of slow growth rate and development of antibiotic tolerance within biofilm [44,45]. This fact was examined by Yogesh and Anjali in 2021 (unpublished data) when they found biofilm-producing *Enterococcus faecalis* resistant to colistin. Despite the full thickness antibiotic penetration, visible cellular activities and protein synthesis have also been seen in oxygen rich areas [46,47]. Through denitrification and fermentation, *P. aeruginosa* can sustain the anaerobic conditions, and supplementation of nitrate or L-arginine can increase the metabolic activity within nutrient deprived areas, thereby increasing the susceptibility to ciprofloxacin and tobramycin [42].

Stringent response (SR) and SOS response are adaptive mechanisms in response to stress and starvation of amino acids, iron, and carbon [48]. The tolerance of *E. coli* for cell wall inhibitor antibiotics such as carbapenems, penicillin, and cephalosporins is believed to be due to SR; this is also thought to be the case for cell division inhibitors such as norfloxacin and ofloxacin [49,50,51,52]. DNA damage may induce the SOS response and can initiate antibiotic tolerance. DNA damage leads to the generation of single-stranded DNA; that may activate RecA, stimulate self-cleavage of the repressor LexA and result in the de-repression of SOS genes [53]. The SOS response may lead to tolerance to antibiotics such as fluoroquinolones that can cause damage to DNA [54]. The tolerance of *E. coli* biofilm to fluoroquinolones has been observed due to the SOS response [52].

### 3.3. Enzymatic Cell Wall Modification

The *dlt* genes are crucial for *Enterococcus faecalis* and *Staphylococcus aureus* to form biofilm [55,56]. This has been observed with the deletion of *S. aureus dltA* and the subsequent reduction of resistance to vancomycin, and also the reduction of planktonic resistance of *E. faecalis* to colistin and polymyxin B [57]. The *dltABCD* operon was a positive hit in a screening for biofilm-specific gentamicin tolerance genes in *Streptococcus mutans*, a dental pathogen that can also cause infective endocarditis [58]. The *dltABCD* homologues are important for D-alanylation of teichoic acid in many Gram-positive species [59]. Due to its inability to add D-alanine to the anionic teichoic acid molecules in the cell wall, the Δ*dltA* mutant was more negatively charged than the wild-type [58]. It is thought that the increased negative charge of the Δ*dltA* strain promotes uptake of gentamicin, a positively charged aminoglycoside [58].

### 3.4. Multispecies Interaction

The interaction between different species (Figure 3) in biofilm can initiate antibiotic tolerance. For example, polymicrobial biofilms of *E. faecalis, Finogoldia magna,* and *S. aureus* were observed to be two-fold more resistant than the mono-species biofilm of *P. aeruginosa* [60]. In the same way, in a dual species biofilm model, *M. catarhhalis* released beta-lactamase, which protected *S. pneumoniae* from amoxicillin [34,61]. In the research by Ryan et al. (2008) on *P. aeruginosa* and *Stenotrophomonas maltophilia* dual-species biofilms, it was observed that the diffusible signal factor is an intercellular signaling molecule produced by *S. maltophilia*, and is sensed by the two-component sensor BptS in *P. aeruginosa*, leading to upregulation of the PmrA-regulated PA3552-3559 and PA4773-4775 genes [62]. Furthermore, the interaction between fungi and bacteria in a multispecies biofilm has also been studied. The resistance of *Staphylococcus* to vancomycin was increased in *C. albicans* and *S. aureus* biofilm due to the fungal matrix component beta-1,3-glucan, which is believed to act as a barrier against vancomycin [63,64]. It was also noted that *C. albicans* can increase the biofilm production of *P. aeruginosa* through alcohol production [65].

### 3.5. Mutation

Genomic mutation, even without any strong spontaneous selective stress or pressure, may lead to AMR. Mutation usually occurs at the rate of 10^−10^ to 10^−9^ per nucleotide per generation in most of the bacteria [66,67]. Oxidative stress-causing agents that damage DNA can also accelerate the mutation rate. Although with a sublethal dose of a bactericidal, the build-up of ROS might be low but would be enough to induce synthesis of multidrug efflux pumps, mutagenesis, and promote resistance [68]. The defect in *mutS*, *mutL*, and *uvrD* genes can further promote the mutation frequency up to 100-fold due to failure of the DNA repair mechanism [69,70]. An example of evolutionary mutation in microorganisms can be seen with hypermutators with the capability to acquire favorable mutations under selective stress that can lead to AMR [71]. This particular phenotype has also been observed with *Pseudomonas* biofilm, with resistance against rifampicin and ciprofloxacin [72]. Apart from that the abovementioned phenotypic characteristics, hypermutations have also been reported in *S. aureus* and *H. influenza* isolated from cystic fibrosis infection but not in *Enterobacteriaceae* isolated from acute urinary tract infection; which indicates hypermutability is favored in certain environments [73,74,75].

Mutation in the *rspL* gene, the gene that codes 16S rDNA and S12 ribosomal protein, affects the antibiotic targeting for aminoglycosides, whereas mutation in the *mexZ* gene results in overproduction of the MexXY-OprM efflux system [76,77]. Additionally, the mutations in the genes coding for the PmrAB two-component regulatory system, which regulates the addition of aminoarabinose to lipid A, has been associated with colistin resistance [78]. Conversely, mutations in the promoter of the chromosomal *ampC* gene that increase the plasmid copy number may result in increased production of β-lactamases [79,80]. A large number of β-lactamase variants with point mutations in the gene, resulting in changes in the amino-acid sequence, may lead to the development of extended-spectrum β-lactamases (ESBLs) that also degrade first-, second-, and third-generation cephalosporins and/or became resistant to β-lactamase inhibitors [81].

### 3.6. Efflux Pump

This is the movement of a drug from the intracellular to extracellular matrix without attachment to an intracellular target (Figure 3 and Figure 4); therefore, this mechanism confers resistance to bacterial cells [82]. Planktonic resistance in *P. aeruginosa* to low concentration ofloxacin has been said to be due to the multiple multidrug efflux pumps, such asMaxAB-OprM [83]. Another major multidrug efflux pump PA1875-1877 is believed to be a contributor to resistance in *P. aeruginosa* biofilm [84]. A two-fold to four-fold increase in susceptibility of biofilm to tobramycin, gentamicin, and ciprofloxacin was observed after the deletion of PA1875, PA1876, and PA1876; on the other hand, planktonic susceptibility was not affected much [84]. In addition, the resistance of *P. aeruginosa* biofilm to azithromycin was said to be due to the MexAB-OprM or MexCD-OprJ efflux pumps, whereas these efflux pumps are said to be required for the resistance against colistin in a metabolically active state [44,85].

### 3.7. Quorum Sensing (QS)

QS is a population-density-dependent regulatory mechanism for interbacterial communication, which acts through signaling molecules named autoinducers. These autoinducers are recognized by the cell-surface receptors in order to induce gene transcription for virulence factors, surface proteins, transcriptional factors, and biofilm production [86,87]. A biofilm formed by *P. aeruginosa* lacking *lasR* and *rhlR* was observed as more susceptible to tobramycin than the wild-type biofilms [88]. In the same way, *S. aureus* deficient with QS-specific *agrD* was observed as less resistant as compared to the wild-type [89]. Moreover, *fsrA* and *gelE* mutants of *E. faecalis* for QS and QS-controlled protease were seen with the impairment of biofilm formation in the presence of daptomycin, gentamicin, or linezolid [90].

## 4. Mechanism of AMR

The four major mechanisms of AMR are illustrated in Figure 4.

### 4.1. Cell Wall or Cell Membrane Modification

Enterobacteriaceae show resistance to carbapenems because of reduced permeability of the bacterial membrane to this antibiotic. In most Enterobacteriaceae, *OmpC* and *OmpF* of *E. coli* are the major porins. The downregulation in the expression of porin or the substitution of major porins with more-selective membrane channels has been engaged in this resistance mechanism [91]. Gram-negative bacteria such as *Pseudomonas aeruginosa*, *V. cholerae,* and *S. enteric* show resistance due to the reduced permeability (Figure 4) of antibiotics like azithromycin, clarithromycin, roxithromycin, and erythromycin. Bacteria can also prevent the access of antibiotic to the target by pumping the antibiotics out of the bacterial cell through efflux pumps. Thus, in Gram-negative bacteria, intrinsic resistance is achieved [92].

### 4.2. Modification or Protection of Targets

Antibiotics are specifically designed to bind to their specific targets with high affinity, and therefore disrupt the normal activity of the target. Antibiotics are less efficient at binding to their targets if the targets alter their structure (Figure 4). A single nucleotide mutation (SNP) in the gene, encoding an antibiotic target, results in resistance towards the given antibiotic. For example, an amino acid substitution in the *rpoB* gene develops resistance towards rifampin. This mutation reduces the affinity of rifampin to its target, and the transcription continues [93].

*Streptococcus pneumoniae* produces the penicillin binding proteins (PBPs), which reduce the affinity to beta-lactam antimicrobial agents. *S. pneumonia is* also resistant to cephalosporin (third-generation drug) due to altered PBP1a and 2x alterations. In PBPs, alteration occurs by the recombination of the PBP gene of *S. pneumonia* and related PBP genes of other closely related streptococcal species by transformation [94]. Without any mutation, antibiotic resistance can be achieved by post-translational modification of targets. In the erythromycin ribosome methylase (*erm*) family, the drug-binding sites alter due to the methylation of 16S rRNA, which therefore prevents the binding of lincosamides, macrolides, and streptogramin to the 16S rRNA [95]. Antibiotics such as lincosamides, phenicols, oxazolidinones, streptogramins, and pleuromutilins may not interact with the target 23S rRNA due to methylation of the A2503 residue by chloramphenicol florfenicol resistance gene (*cfr*) product [96].

### 4.3. Enzymatic Degradation of Antimicrobials

Bacteria can also modify the structure of antibiotics, prevent their entry into the cell, and render them inactive (Figure 4). Bacteria inactivate antibiotics via hydrolysis reactions. Antibiotics such as aminoglycosides, β-lactams, phenicols, and macrolides can be degraded by enzymes such as carbapenemases and chloramphenicol acetyltransferase. The active enzymes against β-lactams include both the early and the extended-spectrum β-lactamases (ESBLs), which are also active against oxyimino-cephalosporins [97]. TEM-1 β-lactamase and SHV-1 (sulphydril variable active site) enzyme, found on the plasmid in a strain of *Escherichia coli*, are the major ESBLs and hydrolyze a broad range of extended spectrum cephalosporins. Changes within antimicrobial functional groups such as thiol, phosphoryl, acyl, or ribosyl due to degrading enzymes may lead to failure of binding of lincomycin, macrolides, and chloramphenicol to targets [98].

### 4.4. Ribosome Protection

Ribosome protection is a resistance mechanism which is developed by some bacteria. Tetracycline is a bacterial protein synthesis inhibitor, and the bacteria produce ribosome protection proteins that bind to the ribosomal target and prevent the binding of tetracycline to the ribosome. In such cases, bacteria can grow even in the presence of tetracycline due to synthesized ribosome protection proteins [99].

## 5. Impact of AMR

The most persistent multidrug-resistant bacteria implicated in high mortality and morbidity globally are *S. aureus*, *Escherichia coli, Enterococcus faecium, Streptococcus pneumoniae, Klebsiella pneumonia,* and *Pseudomonas aeruginosa*. Apart from that, cancer-related neutropenia has been observed with high antimicrobial resistance. Moreover, AMR has complicated the management and treatment of neonatal sepsis due to biofilm formation. Failure of first- and second-generation antibiotics inevitably forces the development of next generation antimicrobials, requiring huge amounts of resources. Resistance of biofilm-producing MDR *Enterococcus faecalis* for a second-line drug (vancomycin) and a last-resort drug (linezolid) was observed by Yogesh and Anjali in 2021 (unpublished data). Morbidity and mortality rates are severely affected by MDR bacterial strains. Without effective antibiotics, organ transplantation, intensive care for pre-term newborns, hip replacement surgery, and chemotherapy for cancer treatment are not usually performed [100].

In the 21st century, biofilm-led AMR has become a global threat to the public health system. Resistant microbes are more difficult to treat, and substitute medications or even higher doses are required to treat them, which may be toxic and more expensive. Microbes resistant to multiple antimicrobials are called multidrug resistant (MDR). All classes of microbes, including bacteria, viruses, fungi, and protozoa, can evolve resistance. Bacteria which are totally drug-resistant (TDR) or extensively drug-resistant (XDR) are called “superbugs” [101]. Bacterial resistance occurs naturally, by genetic mutation, or by procuring genetic material from other bacteria. However, extensive use of antimicrobials appears to encourage selection for mutations that can render antimicrobials ineffective.

## 6. Current Therapeutic Practices against Biofilm Producing Bacteria

Biofilm can be understood by the fact that it is a mechanism of self-motivation in the process of pathogenesis. Numerous techniques and methods have been worked on in order to understand biofilm, target biofilm, and biofilm-producing microorganisms through the medium of natural products, surface coating, lasers, texturing and patterning, nanostructures, and peptides [102,103,104,105]. Figure 5 illustrates a few strategies for alleviating biofilm-related ailments through the means of inhibition, dispersal, or eradication of the biofilm.

### 6.1. Biofilm Inhibition Strategy

The material matrix of implanted medical devices, prosthetic surfaces such as dental filling materials and artificial teeth, and biomaterials provide an ideal site for bacterial adhesion, promoting mature biofilm (Figure 2e,f) formation [106]. Consequently, bacterial adhesion or attachment methods can be worked upon with the aim of preventing biofilm. In doing so, modification of the attachment surface, for example coating the external surface of devices, is an ideal method [107]. Various biomaterials and coating materials which alter the surface of target, making it unfavorable for bacteria, are being developed [106,107]. This has led to the development of coated medical devices for the prevention of biofilm-related infections and/or complications associated with orthopedic implants. Aiming to prevent biofilm, application of therapeutic agents or inhibitors is usually done with orthopedic, dental implants, and dental filling material, such as incorporation of anti-biofilm nanoparticles in dental restorative composite materials; this has resulted in a significant reduction of biofilm, as well as it’s associated infections [106]. Small molecule biofilm inhibitors (Figure 1 and Table 1) are applied to biomaterials and devices as another approach for preventing biofilm formation, as well as to make the surfaces non-reactive or inert. In a clinical trial, 5-fluorouracil was found to be as effective as chlorhexidine/silver sulfadiazine in controlling central venous catheter colonization with no bloodstream infection [108]. Conversely, azithromycin prevented *P. aeruginosa* ventilator-associated pneumonia by targeting quorum sensing in high risk patients [109]. Similarly, azithromycin was seen to improve quality of life without any adverse events in patients with cystic fibrosis infected with *P. aeruginosa* [110]. Natural products such as garlic extracts have also been examined as an inhibitor of quorum sensing; however, no significant effects were detected in patients with garlic therapy as compared with placebo [111]. Already used or utilized inhibitors are therefore becoming the focus of anti-biofilm strategies and research [112].

### 6.2. Dispersal of Biofilm for Treatment

Bacteria in biofilms are inherently more tolerant to antimicrobial treatment when compared directly to planktonic cells (Figure 1) of the same strain [169,170]. The mechanisms of antibiotic resistance in planktonic bacteria, such as mutations, efflux pumps, and antibiotic modifying enzymes, are well understood [93]. These do not hold true for resistance in biofilm formers, as there are many examples where drug-susceptible bacterial strains often exhibit significant antibiotic tolerance in the biofilm; however, they become susceptible once the integrity of biofilm is compromised [39]. Thus, biofilm antibiotic tolerance is thought to involve alternative mechanisms to bacterial antimicrobial resistance.

Quorum sensing, as a principal chemical pathway, enables biofilm-mediated bacterial maintenance and survival [171]. As dispersed cells are generally more susceptible to antimicrobial treatment than biofilm-residing cells, the above strategy has emerged as an intense area of study, resulting in the review, as well as development, of chemical agents capable of effective biofilm dispersal [172,173]. Usually, dispersal agents are employed in parallel with antimicrobials in the case of untreated dispersed cell translocation to different areas and seeding infection [174,175]. Co-treatment generally involves administering a combination of drugs concurrently—in this case, a biofilm dispersal agent and an antibiotic—to exert a synergistic effect [156,174]. Limitation of this treatment strategy can be the availability of both agents at the target site in the correct concentration [176].

### 6.3. Agents to Eradicate Biofilm

A variety of promising candidates, such as antimicrobial peptides (illustrated in Figure 5), capable of targeting and eradicating biofilm, have already been developed; their activity, design, and potential uses have become an emerging field in biofilm research.

Antimicrobial peptides (AMPs) are one of the most well-studied classes of biofilm-eradicating agents, and are often considered an attractive alternative to antibiotics [177,178]. They are composed of 5 to 90 amino acids with a molecular mass of 1 to 5 kDa, and are ubiquitous compounds, produced in a variety of plant, invertebrate, and animal species. While they are mostly cationic in nature, anionic forms have also been reported [80,81]. Despite the intensive research on biofilms, their mode of action is yet to be recognized, except their capabilities of disrupting the cell membrane and their enzymatic or protein activities. While their actions have been observed as a substitute to antimicrobials, AMPs are yet to be recognized as effective agents against biofilms.

Nanoparticles (NPs) are one of the leading drug carriers and are widely researched among all drug delivery systems. They can act as an important tool to breach the biofilm shield and enhance the availability of drugs to the bacterial population (illustrated in Figure 5). Innumerable synthetic polymers are used, along with natural polymers for manufacturing nanoparticles. Inorganic compounds, as well as synthetic and natural polymers, have been used to produce nanoparticles [179]. The most accepted mechanism of action is the ability of NPs to enter the cell in order to induce endocytosis. The authors of this study, in 2021, synthesized silver nanoparticles (AgNPs) (illustrated in Figure 6) of 118–157.7 nm diameter (Figure 7), derived from *Azadirachta indica* (neem), and tested them in various concentrations against multidrug resistant oral pathogen *Enterococcus faecalis*-formed biofilm through a human dentine-block-based biofilm assay (illustrated in Figure 6) and a crystal violet assay. They observed a significant reduction in colony forming units (CFU) (10^7^) upon treatment with AgNPs. Additionally, a significant reduction in CFUs (Figure 8) upon treatment with AgNPs in combination with clove or eugenol was also observed by the authors of this study.

Other proven agents for anti-biofilm properties are chlorhexidine and sodium hypochlorite, which have exhibited a notable reduction in CFU/mL (Figure 8) as compared to AgNPs alone or in combinations with other agents when tested by the authors of this study. It was observed that sodium hypochlorite was as potent as the combined dose of AgNPs and clove. In order to eradicate biofilm-producing pathogens, nanoparticles could be an answer due to their low toxicity, high efficacy power, high and efficient penetration ability into the host cells, and site-specific drug release. Additionally, these properties of NPs can be considered in order to develop an effective method against biofilm as a means to keep AMR at bay.

### 6.4. Bioinformatic Approach to Identify Anti-Biofilm Agent

Accelerated discovery of novel anti-biofilm agents needs new sequencing and computational technologies which are biologically relevant models to better understand biofilm formation. As of now, most studies exploring biofilm mechanisms rely on omics studies, such as transcriptomics and proteomics, to uncover new genetic and protein targets for novel anti-biofilm agents to modulate. Apart from the omics studies, in silico screening can be used to screen for molecules from large databases. Another approach is machine learning, which can predict the anti-biofilm activity of a molecule and can also play a vital role. Candidate molecules thus identified can then be synthesized and validated in several biological models, including biofilms grown in microtiter plates, flow cells, animal models, and human organoids. Therefore, integration of both lab and computational science can provide a better chance of developing a successful anti-biofilm agent.

RNA-Seq, which is a high-throughput technology, is employed to measure gene regulation and expression. It is able to identify transcriptomic signatures highly distinct to biofilms or biofilm-mediated bacterial dispersion [173]. RNA-Seq can also be used to study the effects of antibiotics and potential anti-biofilm agents on biofilm formation [180,181]. Transposon insertion sequencing (Tn-Seq) is a recent technology which offers a high-throughput approach to help in identifying genes required to survive in a specific condition, including in a biofilm. Novel techniques have been identified in Tn-Seq, including sorting mutant cells and probing essential gene functions using inducible promotors [182].

Additionally, for supplementary information, proteomics can be useful, especially meta-proteomics, which can identify proteins essential for multi-species aggregation. Meta-proteomics can also identify the abundance of proteins for each species in key metabolic and energy pathways, such as amino acid metabolism and fermentation absence among communities of single species [183]. Community of four co-cultivated soil bacteria (*Stenotrophomonas rhizophila*, *Xanthomonas retroflexus*, *Microbacterium oxydans*, and *Paenibacillus amylolyticus*) has been seen with increased biofilm formation as compared to single species communities; that suggests a cooperative as well as competitive mechanism of bacterial survival. Metabolomics analyzes differential production of small molecule (Table 1) metabolites and metabolism intermediates (e.g., carbohydrates, nucleotides, and amino acids). Not only that, smaller molecules and intermediates such as amino acids and nucleotides can be analyzed through metabolomics, along with their production. Metabolomics offers a snapshot of the functional changes that result from the transcriptomic and proteomic changes measured by the above methods. Biofilm activity can also be predicted by analyzing metabolites [184,185,186].

## 7. Conclusions

It is a well-established fact that biofilm further complicates the infection associated with communicable, as well as non-communicable, diseases. In addition, biofilm has also been implicated in post-operative infections and digestive disorders such as recurrent sialadenitis. Furthermore, widely studied cystic fibrosis has been identified as a biofilm-associated genetic disorder. Bacteria involved in infective endocarditis are able to form biofilms; that can be associated with the failure of anti-microbials and compromised heart valves. Bacteria capable of producing biofilm have also been identified from atherosclerotic arteries [187]. Surface adherence is one of the prevailing modes for bacterial growth, while the biofilm environment supports the development of antimicrobial resistance [18,188,189]. Our understanding of how this lifestyle influences the evolution of AMR, whether by different population genetic dynamics or molecular mechanisms, is limited. One example is that the close proximity of cells in biofilms may facilitate the horizontal transfer and persistence of resistance genes in bacterial populations [190,191]. There could be many contributing factors assisting the surge in AMR, limiting the use of many antimicrobial agents. Since the role of biofilm in AMR is evident, this consequently opens up avenues for repurposing the existing drugs targeting the biofilm. Forecasting the AMR by analyzing the population of biofilm formers would go a long way to tackle AMR. Secondly, the use of nanoparticles in biomaterials or as a drug delivery mode can be of great use. On the grounds of this, there is compelling requisite for the development of bioinformatics tools with the aim of analyzing, and also predicting, AMR emanating from biofilm.

Furthermore, drug co-administration modalities, as suggested by a number of studies, can precipitate challenges such as complex treatment schedules, increased risk of adverse effects, increased treatment costs, and antagonism [192,193]. Accordingly, in order to predict AMR, in addition to formulating viable therapeutic strategies for biofilm associated ailments, aforementioned facts need to be addressed robustly. Identification of potential protein targets, biofilm formation pathways, and modulators associated with protein targets through in-vitro, and high-throughput virtual screening should be considered. Small molecules, such as 5-fluorouracil, which targets quorum sensing, and synthetic molecules, such as terrain, which targets both quorum sensing as well as c-di-GMP, can be considered for anti-biofilm strategies. Novel molecules, such as SYG-180-2-2, with the ability to reduce the bacterial adhesion and polysaccharide intercellular adhesin (PIA) in methicillin-resistant *Staphylococcus aureus* should be seen as potential antibacterial agents [194,195]. Although the exact anti-biofilm mechanism of a few small molecules or drug candidates such as TAGE and CAGE are not entirely understood, their effectiveness against biofilm-causing microorganisms can still pave the road to combat biofilm-led AMR. Coating of medical devices including implants with silver and zinc oxide nanoparticles is one of the most effective strategies against biofilm-led infection, which can be explored further. Another novel method, machine learning modeling of anti-biofilm agents from a database can also be considered before in-vitro anti-biofilm assays and biological models.

## Figures and Tables

**Figure 1 antibiotics-11-00930-f001:**
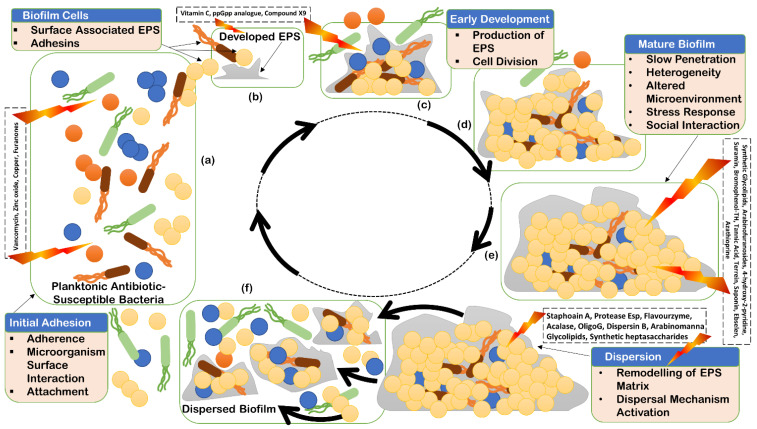
Stages of biofilm life cycle: initial adhesion (**a**); initial developed extracellular polymeric substances (EPS) (**b**); early development of EPS production (**c**); further development of biofilm EPS for maturation (**d**); mature biofilm (**e**); and dispersion of biofilm (**f**).

**Figure 2 antibiotics-11-00930-f002:**
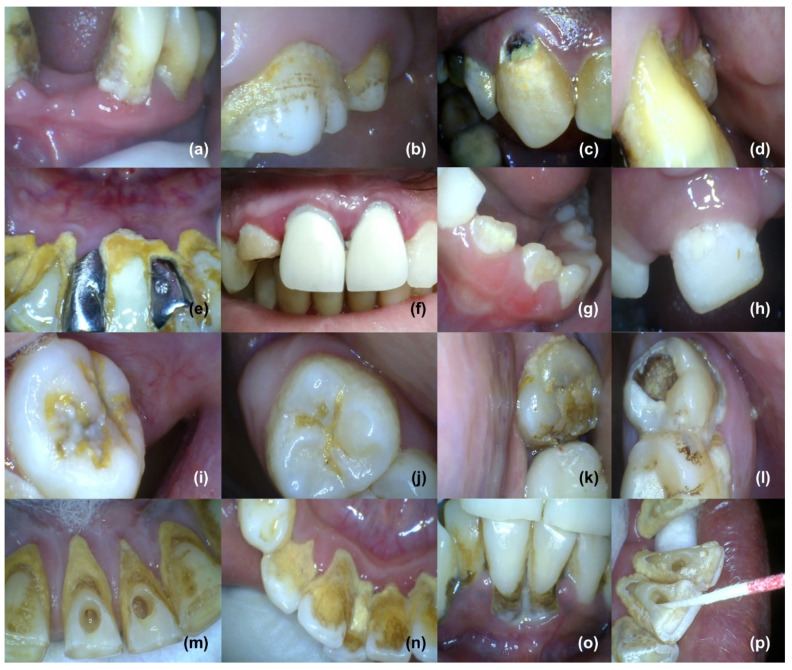
Biofilm on human teeth in the form of plaque and/or calcified plaque (calculus): soft plaque at the neck (near gingiva), crown (more towards gingiva), and more permeable dentogingival junction area of human teeth (**a**–**d**). Dentogingival junction can be seen shifted towards cemento-enamel junction (**a**,**d**), providing more surface for biofilm deposition, exposure of dentine area, and serving as an easy route for passage of bacterial products from biofilm into deep tissues. Hard (calcified) plaque or calculus on artificial surfaces of prosthetic teeth (dental crown or fixed partial denture): more calcified plaque can be seen near the dentogingival junction area, as compared to the crown of prosthetic teeth, and gingival inflammation (gingivitis) of free gingiva may also be seen (**e**,**f**). Plaque on early secondary or permanent, as well as primary, teeth (**g**,**h**). Calcified or hard plaque on the occlusal area forms mainly in pits, fissures, fossae, and grooves of permanent human teeth (**i**,**j**). Change in color of plaque can also be seen, indicating entrapment of minerals from saliva into biofilm and hardening, and soft as well as hard plaque on occlusal, interdental, dentogingival, and cervical areas of permanent human molar teeth (**k**,**l**). Very hard plaque of dark yellow, light yellow, or brown in color near the dentogingival, interdental, and cervical areas of permanent human lower anterior teeth (**m**–**p**). V-shaped shifting of dentogingival junction and free gingiva away from the neck of the teeth, towards the cemento-enamel junction, can also be seen. (Intraoral and dental pictures were provided by Dr. Mamta of Mamta Dental Clinic, Faridabad, India).

**Figure 3 antibiotics-11-00930-f003:**
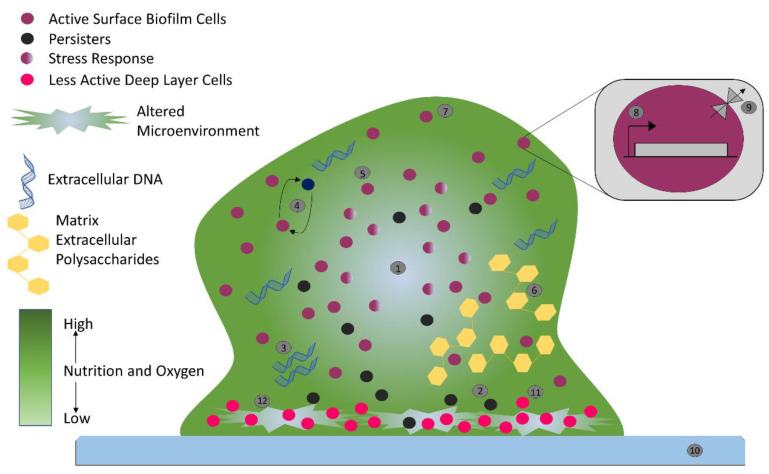
Biofilm-mediated antimicrobial resistance in bacteria: nutrient gradient indicated by color gradient (**1**), persister cells (**2**), extracellular DNA (**3**), intercellular interactions (**4**), stress response (**5**), matrix extracellular polysaccharides (**6**), biofilm cells (**7**), genetic determinants (**8**), multidrug efflux pump (**9**), host surface (**10**), less active deep layer cells (**11**), and altered environment (**12**).

**Figure 4 antibiotics-11-00930-f004:**
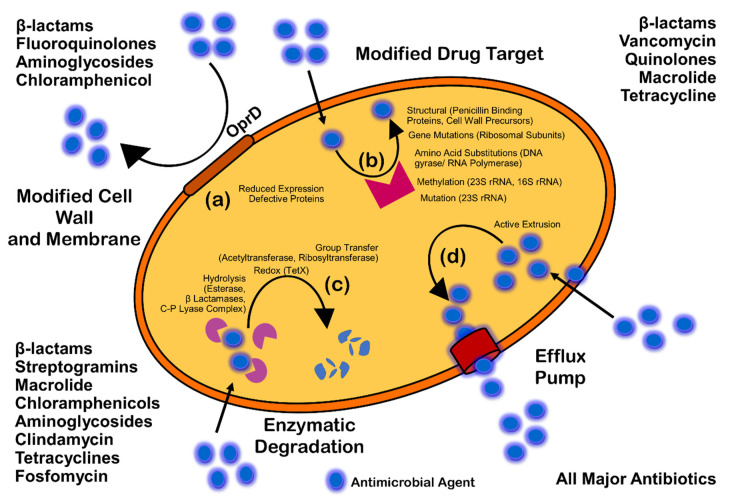
Mechanism of antimicrobial resistance: cell wall modification (**a**); modification of drug target (**b**); enzymatic degradation or destruction of antimicrobials (**c**); and efflux pump (**d**).

**Figure 5 antibiotics-11-00930-f005:**
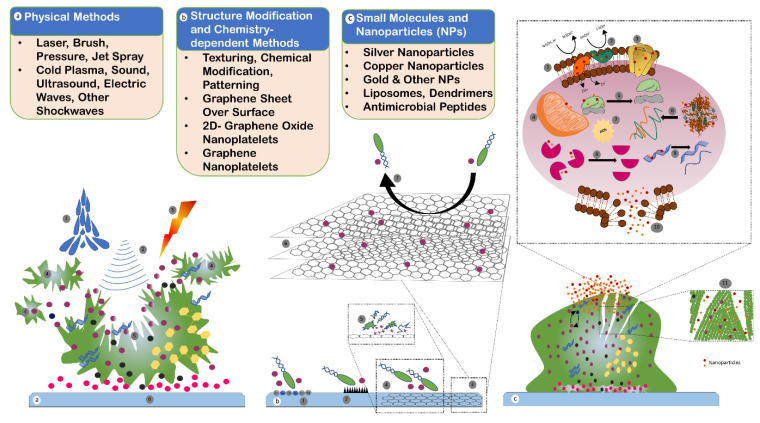
Strategies against biofilm: (**a**) physical methods of biofilm removal include water jet (**1**), sound, ultrasonic, electric and other shockwaves (**2**), and laser and photodynamic therapy (**3**) to dislodge and disrupt the biofilm in small parts (**4**) and to break the main biofilm region (**5**). The host surface or substrate (**6**) is not usually disturbed or modified in this method. (**b**) Structural modification and chemistry-dependent methods include techniques to change the texture, pattern, surface and material properties, surface charge, and chemical coating to repel or avoid bacterial cells attaching to the surface (**1**,**2**). Additionally, graphene sheets on substrate (**3**), graphene nanoplatelets (GNP) (**6**), and chemistry-dependent polyamine-functionalized quantum dots (QDs) are some advanced methods that can be used to inhibit the cell adhesion to substrate (**7**), and to damage the bacterial cells (**4**,**5**) that would provide antimicrobial properties. (**c**) Small molecules and nanoparticles developed and used for biocompatibility and antimicrobial properties include organic nanoparticles (liposomes and polymers), inorganic nanoparticles (silver, copper, gold, and iron oxide nanoparticles), and specifically designed antimicrobial peptides (AMPs) of 5 to 90 amino acids and with a molecular mass of 1 to 5 kDa. AMPs are capable of disrupting bacterial cell membranes, in addition to having enzymatic and protein activities. Nanoparticles can cause damage to the electron transport chain (**1**,**2**), inhibit membrane protein (**3**), dysfunction of the mitochondria (**4**), disassembly of ribosome (**5**), denaturation of proteins (**6**), oxidative stress by producing ROS (**7**), inactivation of enzymes (**8**), damage to DNA (**9**), damage to cell membrane (**10**), and degradation of EPS (**11**).

**Figure 6 antibiotics-11-00930-f006:**
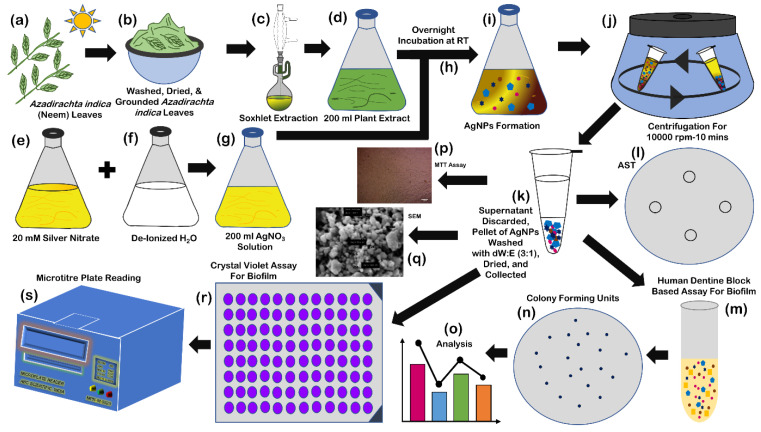
Synthesis of silver nanoparticles: collection of *Azadirachta indica* (neem) leaves (**a**); washing, drying, and grounding of leaves (**b**); Soxhlet extraction (**c**); collection of extract (**d**); preparation of 200 mL 20 mM silver nitrate solution (**e**); 200 mL deionized water (**f**); mixing of silver nitrate solution and deionized water (at the time of mixing) (**g**); incubation at room temperature for 12–18 h (**h**); change of color of mixture after 12–18 h (**i**); centrifugation at 1000 rpm for 10 min (**j**); collection of pellet after discarding the supernatant, washing of pallet with solution of deionized water and ethanol (3:1) for three times, drying, and collection of nanoparticles (**k**); human dentine-block-based biofilm assay (**l**–**o**); cytotoxicity assay (**p**); SEM analysis (**q**); and crystal violet-based biofilm assay (**r**,**s**).

**Figure 7 antibiotics-11-00930-f007:**
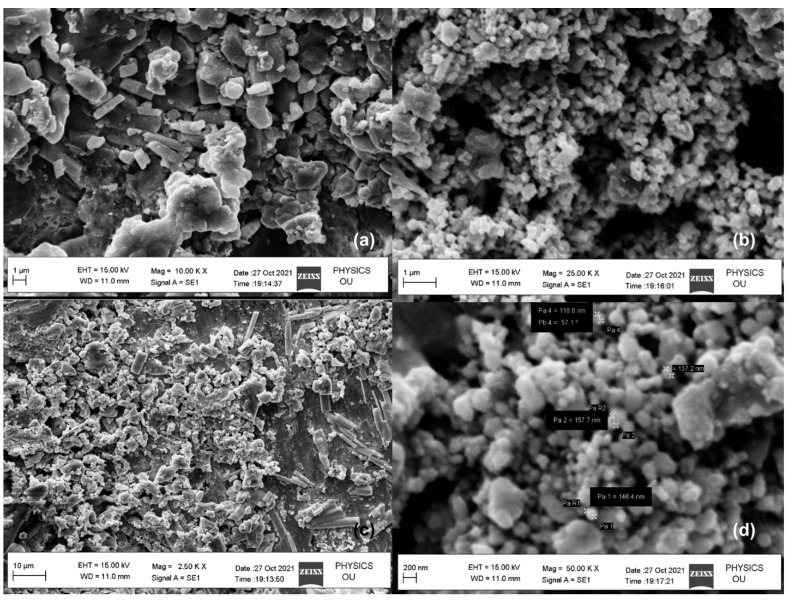
SEM analyses of biosynthesized silver nanoparticles (**a**–**d**).

**Figure 8 antibiotics-11-00930-f008:**
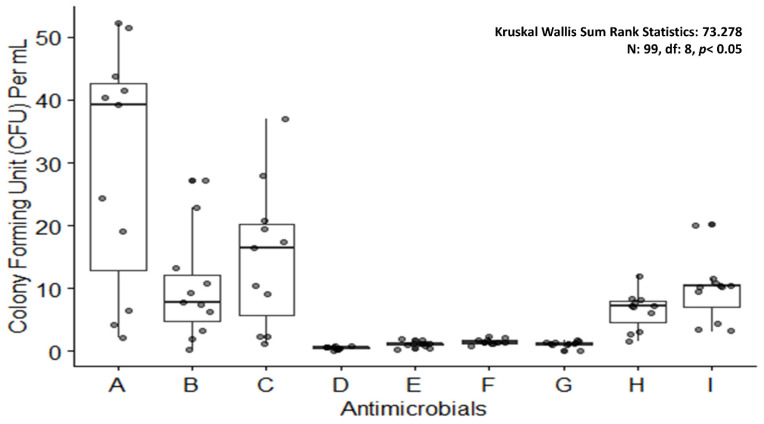
Boxplot showing colony forming units upon treating *Enterococcus faecalis* biofilm with antimicrobials. **A**: control; **B**: silver nanoparticles 20 µg/mL; **C**: silver nanoparticles 10 µg/mL; **D**: silver nanoparticles 20 µg/mL and clove 2500 µg/mL combination; **E**: silver nanoparticles 10 µg/mL and clove 1700 µg/mL combination; **F**: silver nanoparticles 5 µg/mL and eugenol 850 µg/mL combination; **G**: sodium hypochlorite 5%; **H**: sodium hypochlorite 3; **I**: chlorhexidine 2%.

**Table 1 antibiotics-11-00930-t001:** Potential small molecules and drug candidates for biofilm inhibition.

Molecule or Drug Candidate	Target	Reference
3-(trimethoxysilyl)-propyldimethyloctadecyl ammonium chloride (QAS), vancomycin, zinc oxide and silver nanoparticles, iodine, copper, furanone, phloretin, oroidin	Inhibition of bacterial adhesins	[113,114,115]
3,5,6-trideoxy 6-fluorohex-5-enofuranose, terrain, TNRHNPHHLHH, eugenol, azithromycin, 5-fluorouracil (5-FU), benzamide-benzimidazole, penicillic acid, patulin, furanone C30, 5′-methylthio- (MT-), 5′-ethylthio- (EtT-) and 5′-butylthio- (BuT-) DADMe-ImmucillinAs, LMC-21, [*N*-(indole-3-butanoyl)-L-HSL and *N*-(4-bromo-phenylacetanoyl)-L-HSL], S-adenosyl-homocysteine and sinefungin, butyryl SAM, MomL, AiiA, AiiM, *N*-(3-oxododecanoyl)-L-homoserine lactone (3OC12-HSL) and *N*-butanoyl-L-homoserine lactone (C4-HSL), paraoxonases (PON), phenylacetanoyls homoserine lactones (PHLs) and iodo PHLs, phenylbutanoyl HSL (PBHL), licoricone, glycyrin, glyzarin and flavan-3-ol catechin, ajoene (flavonoids from garlic), iberin, cheirolin, iberverin, sulforaphane, alyssin, erucin, ursolic acid, ferric ammonium citrate (FAC), compound 59, baicalein, palmitoyl-DL-carnitine (pDLC), palmitic acid, carnitine, patulin, TP-1, salicylic acid, nifuroxazide, chlorzoxazone, N-(heptylsulfanylacetyl)-l-homoserine lactone, 3-alkyl-5-methylene-2(5*H*)-furanones (HFs), (5Z)-4-bromo-5-(bromomethylene)-3-butyl-2(5H)-furanone, oroidin and bromoageliferin, bis(deacetyl)solenolide D, ethyl *N*-(2-phenylethyl)carbamate, *N*,*N*-dichloro, isocyanide, isothiocyanate, dithiocarbamate derivatives of 2-(4-nitrophenyl)ethylamine, benzoic acid, aeroplysinin-I, bromoageliferin, 5-methoxy2-[(4-methyl-benzyl)sulfanyl]-1H-benzimidazole (ABC-1), N-ethyl-3-amino-5-oxo-4-phenyl-2,5- dihydro-1H-pyrazole-1-carbothioamide, allicin (diallylthiosulphinate), brominated furones, and amburic acid	QS/AHL/AI/AHL-acylase	[108,116,117,118,119,120,121,122,123,124,125,126,127,128,129,130,131,132,133,134,135,136,137,138,139,140,141,142,143,144,145,146,147]
GSK- X9, terrain, saponin, vitamin C, sulfathiazole and azathioprine, LP 3134, LP 3145, LP 4010 and LP 1062, Amb2250085 and Amb379455, ebselen (Eb) and ebselen oxide (EbO), benzoisothiazolinone derivative, H19 and 925 (hiol-benzo-triazolo-quinazolinones), palmitic acid, and palmitoyl-dl-carnitine (pdlc)	Nucleotide second messenger signaling systems/ second messenger cyclic dimeric guanosine monophosphate/ guanosine diphosphate (GDP) guanosine tetraphosphate (ppGpp), guanosine pentaphosphate (pppGpp), bis(3′,5′)-cyclic diguanylic acid (c-di-GMP), c-di-AMP, Rel enzyme, DGC	[118,137,148,149,150,151,152,153,154,155]
Azathioprine, ebselen, sulfonohydrazide, sodium nitroprusside (SNP), S-nitroso-L-glutathione (GSNO), and S-nitroso-N-acetylpenicillamine (SNAP)	Diguanylate cyclase enzymes (DGCs)	[113,150,156]
Bromoageliferin, TAGE (*trans*-bromoageliferin) and CAGE (*cis*-bromoageliferin), amburic acid, and 4-epi-pimaric	Unspecific	[113,157,158,159]
Biphenylmannosides and dihydrothiazolo ring-fused 2-pyridone scaffold, bicyclic 2-pyridone, tetrazoles, acyl sulfonamides and hydroxamic acids (Mannocides/Pilicides), bicyclic b-lactams, dihydroimidazolo, and monocyclic 2-pyridone	Biofilm formation/chaperone	[158,160,161,162,163,164,165]
Q24DA	Motility	[166]
Naringenin, quercetin, and polyphenol ellagic acid	AI-2-mediated cell–cell signaling	[167,168]
5-methoxy2-[(4-methyl-benzyl)sulfanyl]-1H-benzimidazole (ABC-1)	SpA, PIA, eDNA	[136]

## Data Availability

The data presented in this study are available on request from the corresponding author. The data are not publicly available they have not been added to any repository yet.

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
