# Peer review of "The Association between Biofilm Formation and Antimicrobial Resistance with Possible Ingenious Bio-Remedial Approaches"

_antibiotics, 2022, doi:10.3390/antibiotics11070930_

Round 1

Reviewer 1 Report

Dear Authors,

The paper "The Association between Biofilm Formation and Antimicrobial Resistance with Possible Ingenious Bio-Remedial Approaches", presents a complex problem of the development of antimicrobial resistance in biofilms.

The understanding of the molecular mechanisms is still limited, but not unknown.

There are many studies and reviews on AMR topics as presented here as references.

It is to be appreciated that the novel approach with respect to machine learning modelling of all agents for and against biofilm in the database to identify an effective model targeting biofilm can be considered before in-vitro anti-biofilm assay and biological modelling.

In other words, I recommend the publication of this review, which makes a contribution to the field of association between biofilms and antimicrobial resistance. The paper is well organized and comprehensively described, and the references are topical and appropriate with previous related work. 

Author Response

Response to Reviewer 1 Comments:

Point 1: The paper “The Association between Biofilm Formation and Antimicrobial Resistance with Possible Ingenious Bio-Remedial Approaches”, presents a complex problem of the development of antimicrobial resistance in biofilms.

Response 1: Thank you for the kind review.

Point 2: The understanding of the molecular mechanisms is still limited, but not unknown.

Response 2: Thank you for the kind review. We have attempted to address this point.

Point 3: There are many studies and reviews on AMR topics as presented here as references.

Response 3: Thank you for the kind review. We have referred studies covering different areas of AMR like studies on the biofilm and biofilm-led AMR.

Point 4: It is to be appreciated that the novel approach with respect to machine learning modelling of all agents for and against biofilm in the database to identify an effective model targeting biofilm can be considered before in-vitro anti-biofilm assay and biological modelling.

Response 4: Thank you for the kind review. We believe this approach of machine learning modelling would help in designing strategies against biofilm. 

Point 5: In other words, I recommend the publication of this review, which makes a contribution to the field of association between biofilms and antimicrobial resistance. The paper is well organized and comprehensively described, and the references are topical and appropriate with previous related work.

Response 5: Thank you for the kind review.

Reviewer 2 Report

The review article covers an interesting topic related to the infectious disease. However, the manuscript should be revised before considering it for the publication. Below some recommendation for improving the quality of the manuscript.

-English grammar should be rigorously checked since there are  a lot of typos errors and grammar errors that make the reading very difficult. A revision by a native speaker is necessary.

-considering the sections in the review. I recommend to introduce a picture and/or a Table reporting the most promising small-molecules developed for targeting biofilm formation. Authors should also report clinical trials about the possible drug candidates. In my opinion, this aspect should be treated according to the journal, given that the title is Antibiotics. Considering that in the paper there are no mention about molecules/drug candidates, a revision in this sense is necessary.

-Second point, the authors should report a perspective about this argument and an outlook on the future to improve the conclusion section.

Author Response

Responses:

Point 1: The review article covers an interesting topic related to the infectious disease. However, the manuscript should be revised before considering it for the publication. Below some recommendation for improving the quality of the manuscript.

Response 1: Thank you for the kind review. The corrections have been done.

Point 2: -English grammar should be rigorously checked since there are a lot of typos errors and grammar errors that make the reading very difficult. A revision by a native speaker is necessary.

Response 2: Thank you for the kind review. The corrections have been done.

Point 3: -considering the sections in the review. I recommend to introduce a picture and/or a Table reporting the most promising small-molecules developed for targeting biofilm formation. Authors should also report clinical trials about the possible drug candidates. In my opinion, this aspect should be treated according to the journal, given that the title is Antibiotics. Considering that in the paper there are no mention about molecules/drug candidates, a revision in this sense is necessary.

Response 3: Thank you for the kind review. The corrections have been done. The figure 1 has been edited to show small molecules targeting at particular stage of life-cycle of biofilm (e.g. early stage, mature or dispersion); also, a supplementary table reporting the small-molecules for targeting biofilm has also been added. We have also reported clinical trials about the drug candidate in section 5.1.

Point 4: -Second point, the authors should report a perspective about this argument and an outlook on the future to improve the conclusion section.

Response 4: Thank you for the kind review. A perspective and an outlook on the future have been added/ mentioned in the conclusion section.

Reviewer 3 Report

The content and figures presented in the article are commendable but it is really difficult to read the article due to lots of mistakes in grammar and sentence construction. I strongly suggest that this article must go through proofreading by a English language professional before publication.  

Some of the examples of mistakes are given below but similar mistakes are spread throughout the paper:

Line 77: To convey the meaning correctly, the title should read “Overview of Biofilm-led survival against antibiotics”

Line 78-80: Sentence construction is very complicated. It should be re-written and if necessary be divided into multiple sentences.

Line 85: wrong English. Should be written as “……and are embedded in”

Line 86, 87: The entire sentence should be re-written. There are many mistakes. For example: “cannot be predicted” is the correct use.  

Line 88: “These are” should be replaced with “Biofilms are” at the beginning of sentence.

Line 90: “colonize” is correct use. No need to use past tense here.

Line 95-96: Correct use is “….and they worsen”

Line 114: the superscripts on the digits and g (gram) look very small and not readable.

Author Response

Responses:

Point 1: The content and figures presented in the article are commendable but it is really difficult to read the article due to lots of mistakes in grammar and sentence construction. I strongly suggest that this article must go through proofreading by a English language professional before publication.  

Response 1: Thank you for the kind review. The corrections have been done.

Point 2: Line 77: To convey the meaning correctly, the title should read “Overview of Biofilm-led survival against antibiotics”

Response 2: Thank you for the kind review. The sentence has been corrected.

Point 3: Line 78-80: Sentence construction is very complicated. It should be re-written and if necessary be divided into multiple sentences.

Response 3: Thank you for the kind review. The sentence has been corrected.

Point 4: Line 85: wrong English. Should be written as “……and are embedded in”

Response 4:  The sentence has been corrected.

Point 5: Line 86, 87: The entire sentence should be re-written. There are many mistakes. For example: “cannot be predicted” is the correct use.  

Response 5:  The sentence has been corrected.

Point 6: Line 88: “These are” should be replaced with “Biofilms are” at the beginning of sentence.

Response 6: The sentence has been corrected.

Point 7: Line 90: “colonize” is correct use. No need to use past tense here.

Response 7: The sentence has been corrected.

Point 8: Line 95-96: Correct use is “….and they worsen”

Response 8: The sentence has been corrected.

Point 9: Line 114: the superscripts on the digits and g (gram) look very small and not readable.

Response 9: Thank you for the kind review. The sentence has been corrected.

Reviewer 4 Report

Dear Author! 

Interesting scientific work. Accept! 

Author Response

Response:

Point 1: Interesting scientific work. Accept!

Response 1: Thank you for the kind review.

Round 2

Reviewer 2 Report

My concerns were addressed in the revised version.